# Linkage of Maternity Hospital Episode Statistics data to birth registration and notification records for births in England 2005–2014: Quality assurance of linkage of routine data for singleton and multiple births

Gillian Harper

► http://dx.doi.org/10.1136/bmjopen-2017-017897

Centre for Maternal and Child Health Research, University of London, London, UK

**Correspondence to**
Dr Gillian Harper;
gill.harper@city.ac.uk

## ABSTRACT

**Objectives** To quality assure a Trusted Third Party linked data set to prepare it for analysis.

**Setting** Birth registration and notification records from the Office for National Statistics for all births in England 2005–2014 linked to Maternity Hospital Episode Statistics (HES) delivery records by NHS Digital using mothers' identifiers.

**Participants** All 6 676 912 births that occurred in England from 1 January 2005 to 31 December 2014.

**Primary and secondary outcome measures** Every link between a registered birth and an HES delivery record for the study period was categorised as either the same baby or a different baby to the same mother, or as a wrong link, by comparing common baby data items and valid values in key fields with stepwise deterministic rules. Rates of preserved and discarded links were calculated and which features were more common in each group were assessed.

**Results** Ninety-eight per cent of births originally linked to HES were left with one preserved link. The majority of discarded links were due to duplicate HES delivery records. Of the 4854 discarded links categorised as wrong links, clerical checks found 85% were false-positives links, 13% were quality assurance false negatives and 2% were undeterminable. Births linked using a less reliable stage of the linkage algorithm, births at home and in the London region, and with birth weight or gestational age values missing in HES were more likely to have all links discarded.

**Conclusions** Linkage error, data quality issues, and false negatives in the quality assurance procedure were uncovered. The procedure could be improved by allowing for transposition in date fields, and more discrimination between missing and differing values. The availability of identifiers in the datasets supported clerical checking. Other research using Trusted Third Party linkage should not assume the linked dataset is error-free or optimised for their analysis, and allow sufficient resources for this.

## INTRODUCTION

The quality assurance procedure described here was developed to support a project[1] for which data recorded at birth registration, birth

### Strengths and limitations of this study

► The study is the first time linked birth registration and hospital delivery records for 10 years of data at the national level have been quality assured.
► A high linkage rate was maintained after quality assurance.
► The methodology was able to draw upon the availability of identifiers in both the Office for National Statistics (ONS) and Hospital Episode Statistics (HES) data to inform the choice of rules at each stage in the methodology, to carry out clerical checks to ensure the accuracy of the results and to explore data and linkage quality issues.
► Carrying the work out in the secure environment of ONS' Virtual Microdata Laboratory safeguarded the data but limited software options and increased processing time.
► Any bias arising from non-submission of HES records by some trusts is likely to lead to bias in the extent to which records could be quality assured.

notification and clinical information on maternity care at delivery were linked for analysis. After piloting in an earlier project,[2] the Office for National Statistics (ONS) mainstreamed the linkage of data recorded at birth registration in England and Wales to data recorded at birth notification, originally in an interim system, NHS Numbers for Babies. These linked records of births from 2005 to 2014 were then linked to records from the Maternity Hospital Episode Statistics (HES) data set for births that occurred in England and from the National Community Child Health Database/Patient Episode Database Wales for births that occurred in Wales. This linkage brought together key demographic and clinical data items unique to each data set and not otherwise available together at a national level, enabling new

**BMJ**

analyses. For example, it made it possible to compare time of birth, recorded in birth notifications with birth outcomes such as mortality and conditions reported at birth, as ONS routinely links child death registration to birth registration records and reports. The further linkage with Maternity HES made it possible to analyse the outcomes of birth by onset of labour, mode of delivery, gestational age, time of day and day of the week. The history of the project and details of the linkage are described elsewhere.[3]

The routinely linked ONS birth registration and birth notification records from here on will be referred to collectively as ONS birth records.

This paper describes quality assurance of the linked data for England, as data for Wales were linked separately.

Quality assuring this linkage aims to achieve two goals simultaneously. The first is to ensure that the ONS birth record has linked to the correct HES delivery record. As HES can contain more than one episode record for a delivery, the second aim of quality assurance is to select the most relevant HES delivery episode record with the most information from all the available correctly linked records.

### Data sources

The ONS birth records were for 6 468 586 singleton and 208 326 multiple births in England from 1 January 2005 to 31 December 2014.

Only the Maternity HES delivery record which contains information on onset of labour and method of delivery was used. Maternity HES also contains a birth record for the baby, but linkage to these data and the further linkage to subsequent admissions of mothers and babies are not considered in this paper.

HES delivery records are usually grouped by financial year and were provided for the financial years 1 April 2004 to 31 March 2015 in order to include the calendar years used by ONS for its publications and analyses. They are for finished consultant episodes which capture information about diagnoses and procedures during the time a patient spends under the care of an individual consultant, or in the case of maternity, a midwife. Each delivery record contains an admitted patient care record with additional fields about the mother in an appended maternity tail, and information on the baby in an appended baby tail for up to nine babies.

Singleton and multiple birth deliveries are not explicitly identified in HES, but multiple delivery records can be identified where the diagnostic codes and the number of babies indicate a multiple birth, and vice versa for singletons. On this basis, 7 040 590 HES delivery records relating to singleton births, and 123 497 relating to multiple births were identified. As stated above, the singleton and multiple births were quality assured separately. There can be more than one episode record per delivery, and some records are duplicated, therefore the number of HES delivery records is higher than the number of births.

Full details of the ONS and HES data are given elsewhere[3 4] along with assessments of their data quality.[5 6] HES, particularly Maternity HES, has well recognised data quality problems[7] such as large numbers of missing data for some variables and missing maternity tails. The data recorded at birth registration are overseen by the General Register Office and processed by ONS for statistical purposes. ONS official statistics are considered as a 'gold' standard. Data quality has been found to influence the accuracy of linkage and create bias.[2 5 6 8 9] Methods of cleaning HES data in isolation and by trust are discussed elsewhere.[10–13]

### Linkage

Linkage of the ONS birth records to the Maternity HES delivery records was carried out by the Health and Social Care Information Centre, now known as NHS Digital, at the person level, using maternal identifiers because HES delivery records are associated with mothers rather than their babies. It consists of a hierarchical stepwise linkage based on combinations of NHS number, date of birth, sex and postcode of the mother's place of residence (see table 1). Every link is assigned the step of the algorithm that made the link, known as the match rank. On this basis, registered births to a mother during the study period 2005–2014 should link to all available delivery records for every birth that mother has had in those 10 years.

Mothers' identifiers from ONS birth records were linked first of all to the HES Patient Index to assign a HESID to each. The HESID is a pseudonymised number which uniquely identifies a patient and provides a way of tracking them in HES.[14] Then the HES delivery records associated with these HESIDs were extracted. Table 2A,B gives a summary of the linkage rates by calendar year, showing that 97% of singleton births and 95% of multiple births were linked to at least one HES delivery record.

Instances where the linkage algorithm did not link a registered birth to an HES delivery record could be because no delivery record existed, for example, for home births and births in private health facilities which are not usually captured by HES, non-submission of HES data by particular trusts at particular times, or due to linkage error. Overall, 1.7% of singleton and 4.3% of multiple ONS birth records without a link to an HES delivery record did have a link to the mother's HESID. These problems have not been investigated further in this paper.

This paper focuses only on validating the records that were linked to a mother's HESID and HES delivery record by NHS Digital's algorithm. Singleton and multiple births were assessed separately due to the extra complexity associated with multiple birth registration records. This paper sets out the quality assurance procedures and results for both.

**Table 1** NHS Digital hierarchical stepwise linkage algorithm used to link ONS birth records to HES delivery records

| Step/match rank | NHS number | DOB | Sex | Postcode | |
|---|---|---|---|---|---|
| 1 | Exact | Exact | Exact | Exact | |
| 2 | Exact | Exact | Exact | | |
| 3 | Exact | Partial | Exact | Exact | |
| 4 | Exact | Partial | Exact | | |
| 5 | Exact | | | Exact | |
| 6 | | Exact | Exact | Exact | Where NHSNO does not contradict the match and DOB is not 1 January and the postcode is not in the 'ignore' list |
| 7 | | Exact | Exact | Exact | Where NHSNO does not contradict the match and DOB is not 1 January |
| 8 | Exact | | | | |

DOB, date of birth; HES, Hospital Episode Statistics; NHSNO, NHS number; ONS, Office for National Statistics.

## METHODS

The aim was to assess the linked ONS birth and HES delivery records at the national level to efficiently identify one correctly linked HES record with the maximum amount of delivery information from all possible links to create a file to be used in analysis.

Both mothers and babies' identifiers were available for the ONS birth and HES delivery records to use in the quality assurance process. These were mother's date of birth, NHS number and postcode of usual residence, and baby date of birth, NHS number and sex.

The work was carried out in the secure environment of ONS' Virtual Microdata Laboratory (VML). Microsoft Access database software was used as it was the only database software available in the VML at that time, and Structured Query Language (SQL) code was used for replicability. The relatively large size of the data files and restrictions on database size and processing speed constrained the design of the methods used.

The REporting of studies Conducted using Observational Routinely-collected Data (RECORD)[15] guidelines and checklist were followed where applicable.

### Data preparation

It was necessary to normalise the baby information in the maternity tail section of each HES delivery record so that the data could be stored and interrogated efficiently in the database environment. This meant that if information was held on more than one baby in the maternity tail part of a delivery record, this was transformed into separate records for each baby. For example, this meant two records, one for each of the two babies in a twin delivery. Each record contained the same common episode identifier, admitted patient care fields and maternity tail fields, but with the respective baby's information in each one.

For variables common to birth registration and birth notification, the maximum amount of reliable information was derived from the two sources. The baby's date of birth, birth weight and sex were taken from birth registration, and gestational age was taken from birth notification, unless values were missing.

Data types and formats, for example, date fields and values for the baby's sex, were made consistent between ONS and HES records.

### Data quality

The extent of missing values for key items of data about the baby in the ONS birth records for 1 January 2005 to 31 December 2014 and HES normalised records for 1 April 2004 to 31 March 2015 is given in table 3. A value would have to be missing on both birth registration and birth notification to be considered missing overall on ONS birth records in table 3. While the total numbers may not be directly comparable due to definitional and date differences, the percentages of missing values per variable give an indication of differences in data completeness.

The HES delivery data have a much higher proportion of records with missing values than ONS birth records, for example, up to 37% for gestational age in singleton births. The proportion of records with missing information is lower for multiple births in HES, as data items for the first baby are often repeated for their siblings in the same delivery.

Where the baby's date of birth was missing on HES, it was derived from operation dates if a delivery code was present. If sex of baby, gestational age or birth weight was missing, they could not be derived, however.

Gestational age and birth weight are known to have problems, notably implausible values, truncation and rounding in HES. ONS birth records can also have implausible values for birth weight but this has been improving with the introduction of 'warning' edits to birth notification systems.

### Procedure

There are few examples of existing procedures to quality assure linked data sets. One example[16] cleaned and validated linked HES and mortality data for England by reformatting and deriving variables, removing duplicates, and breaking erroneous links indicated where common data items were highly discordant.

**Table 2** Summary of linkage rates for singleton and multiple births by calendar year

**(A) Singleton births by calendar year**

| Year | Total singleton ONS birth records | Total singleton ONS births linked to any Maternity HES by NHS Digital | Total singleton ONS births not linked to any Maternity HES by NHS Digital | Singleton ONS birth records linked to HESID but no delivery record | % Total singleton ONS births linked to any Maternity HES by NHS Digital |
|---|---|---|---|---|---|
| 2005 | 599 237 | 565 559 | 33 678 | 16 349 | 94.4 |
| 2006 | 620 730 | 589 127 | 31 603 | 15 802 | 94.9 |
| 2007 | 638 995 | 612 782 | 26 213 | 12 889 | 95.9 |
| 2008 | 656 196 | 635 411 | 20 785 | 9908 | 96.8 |
| 2009 | 653 322 | 636 284 | 17 038 | 9099 | 97.4 |
| 2010 | 665 599 | 652 533 | 13 066 | 7009 | 98.0 |
| 2011 | 666 582 | 653 552 | 13 030 | 8329 | 98.0 |
| 2012 | 676 399 | 661 511 | 14 888 | 11 444 | 97.8 |
| 2013 | 647 666 | 633 222 | 14 444 | 11 913 | 97.8 |
| 2014 | 643 860 | 628 032 | 15 828 | 13 514 | 97.5 |
| Total | 6 468 586 | 6 268 013 | 200 573 | 116 256 | 96.9 |

**(B) Multiple births by calendar year**

| Year | Total multiple ONS birth records | Total multiple ONS births linked to any Maternity HES by NHS Digital | Total multiple ONS births not linked to any Maternity HES by NHS Digital | Multiple ONS birth records linked to HESID but no delivery record | % Total multiple ONS births linked to any Maternity HES by NHS Digital |
|---|---|---|---|---|---|
| 2005 | 18 376 | 17 404 | 972 | 510 | 94.7 |
| 2006 | 19 541 | 18 522 | 1019 | 602 | 94.8 |
| 2007 | 20 066 | 19 257 | 809 | 450 | 96.0 |
| 2008 | 20 803 | 20 100 | 703 | 378 | 96.6 |
| 2009 | 22 008 | 21 338 | 670 | 369 | 97.0 |
| 2010 | 21 501 | 21 033 | 468 | 276 | 97.8 |
| 2011 | 22 099 | 21 199 | 900 | 722 | 95.9 |
| 2012 | 22 058 | 20 166 | 1892 | 1805 | 91.4 |
| 2013 | 20 767 | 18 735 | 2032 | 1917 | 90.2 |
| 2014 | 21 107 | 19 015 | 2092 | 2009 | 90.1 |
| Total | 208 326 | 196 769 | 11 557 | 9038 | 94.5 |

HES, Hospital Episode Statistics; ONS, Office for National Statistics.

A pilot was initially carried out on data for 2005 to help formulate the methodology. The following procedure is the final version that was carried out on all 10 years of the linked data. Clerical checks were carried out throughout the development phase using relevant variables and identifiers to inform and improve the procedure.

**Table 3** Comparison of percentages of missing values in key baby data items on ONS birth records and HES normalised delivery records for singleton and multiple births 2005–2014

| Variable | ONS singleton birth records | HES singleton delivery records | ONS multiple birth records | HES multiple delivery records |
|---|---|---|---|---|
| Total | 6468 586 | 7040 590 | 208 326 | 230 019 |
| % with missing values for: | | | | |
| Baby's date of birth | 0.00 | 19.4 | 0.00 | 16.3 |
| Sex of baby | 0.00 | 23.1 | 0.00 | 16.1 |
| Gestational age | 0.71 | 37.2 | 0.80 | 27.8 |
| Birth weight | 0.56 | 22.7 | 1.70 | 15.3 |

HES, Hospital Episode Statistics; ONS, Office for National Statistics.

**Table 4** Categories of linked ONS birth records and HES delivery records

| Category | Summary | Details | Action |
|---|---|---|---|
| SMSB | ONS birth record linking to one relevant HES delivery record | One-to-one linkage as expected | Keep link |
| SMSB | ONS birth record linking to many relevant HES delivery records, no clear episode order sequence | Due to duplicate HES records, or predelivery and postdelivery HES records | Keep link to one of the duplicates or the most appropriate delivery record Move others to unlinked file |
| SMSB | ONS birth record linking to many relevant HES records, with a clear episode order sequence | Due to multiple episodes as part of a hospital spell | Keep link to one episode record and maximise delivery information Move others to unlinked file |
| SMDB | ONS birth record linking to one or more incorrect HES delivery records (same mother) | Another birth to the same mother within the study period | Move to unlinked file |
| WL | ONS birth record linking to one or more incorrect HES delivery records | The maternal linkage is incorrect or poor data quality | Move to unlinked file |

HES, Hospital Episode Statistics; ONS, Office for National Statistics; SMDB, same mother different baby; SMSB, same mother same baby; WL, wrong link.

On the assumption that the maternal linkage was correct, the links between the ONS birth records and HES delivery records were categorised as either 'same mother same baby (SMSB) or 'same mother different baby' (SMDB). The category 'wrong link' (WL) was for where neither of the previous categories apply, suggesting incorrect maternal linkage or data quality problems. These scenarios are summarised in table 4. This excludes occasions when a birth registration record did not link to any HES delivery records.

An ONS birth record may link to one correct or incorrect HES delivery record, or to more than one HES delivery record with one, none or some of them being correct for any of the reasons in table 4.

The quality assurance exercise aimed to keep 'same mother same baby' linked records (with the correct normalised record for multiple births), and move all others to the unlinked file. If more than one correct link was available, only one linked delivery record containing the maximum delivery information was chosen. This was to create a one-to-one relationship and support ease of analysis.

Baby data items were compared to ascertain if the linked records related to the same baby. These were baby's date of birth (derived date of birth in HES where applicable), sex, gestational age, birth weight and location of birth.

The place of birth codes used differed between the data sets. These were translated into a consistent common location code, designed specifically for the study.[17] This accounted for NHS organisational changes which took place over the time period. Many maternity services continued to operate in the same location, while the NHS trusts responsible for them and, consequently their organisation code, changed over the study period. In addition, groups of location codes were identified that were within a reasonable geographical distance of each other that might transfer women between them, for example, if they developed problems which made it advisable to transfer to a hospital with a specialist neonatal unit. The presence of codes within these groups on the linked data could relate to the same birth.

Locations of births out-with an NHS trust, such as home births, were all treated as potential location matches because they could ultimately be transferred to an NHS hospital and have an HES delivery record, and the recording of the location for such births in HES can be unreliable. For these cases, the other baby data items were relied on to establish if records were for the same baby.

Four combinations of these variables were used to confirm whether or not a linked ONS birth record and HES delivery record related to the same baby (see table 5), in order of decreasing certainty from 1 to 4. Combination 1 required all the data items to match. Combination 2 required location and date of birth to match plus either birth weight or gestational age or sex. Combination 3 required only location and date of birth to match and birth weight, gestational age and sex will not match in these cases. Combinations 2 and 3 took into account the high occurrence of missing values and poor data quality in HES. Combination 4 accounted for a degree of data entry error in the date of birth by allowing it to differ by up to 4 days, but only if backed up by the location of birth matching, and birth weight and gestational age and sex not differing very much between ONS and HES if the values were not missing. If they were different and the date of birth was not an exact match, it was considered to be a different baby. The four combinations were applied in order so the most reliable matches are identified first.

If the linked records referred to the same baby and there was more than one linked HES delivery record to choose from, then the record with the maximum onset of delivery and delivery method information was chosen, as these variables were of key importance for the analysis. The presence of valid values in these and other fields was used to inform this.

**Table 5** The four combinations of baby data items that confirm that the link between the ONS birth record and HES relates to the same baby

| Combination | ONS birth record and HES delivery location of birth matches | | ONS birth record and HES delivery baby date of birth matches | | ONS birth record and HES delivery birth weight matches | | ONS birth record and HES delivery gestational age matches | | ONS birth record and HES delivery sex of baby matches |
|---|---|---|---|---|---|---|---|---|---|
| 1 | Exact | and | Exact | and | Exact | and | Exact | and | Exact |
| 2 | Exact | and | Exact | and | (Exact | or | Exact | or | Exact) |
| 3 | Exact | and | Exact | and | | | | | |
| 4 | Exact | and | Differs by up to 4 days* | and | Missing or not completely different | and | Missing or not completely different | and | Missing or not completely different |

*By up to 10 days for multiple births.
HES, Hospital Episode Statistics; ONS, Office for National Statistics.

The procedure was implemented using deterministic stepwise rules to be carried out in order, with any linked records dealt with at each stage excluded from subsequent stages. A summary is given in box 1 and fuller details are given in online supplementary appendix A (supplementary appendix A table 1 for singleton births, and supplementary appendix A table 2 for multiple births). The complexity of quality assurance was greater for multiple births because each ONS birth record linked to all HES normalised records for that delivery through the mother's identifiers, by corresponding to each of the siblings. The procedure is slightly different for singleton and multiple births to account for this. For example, for multiple births, the birth order is taken into consideration.

The first two rules cleaned the HES data by eliminating invalid linked records and duplicates. It is known that some trusts generate multiple copies of the same delivery episode record, usually six or nine copies of the same record. In these cases, only one was kept to be quality assured and the others were discarded for efficiency.

### Checks
Once all rules in the procedure had been carried out, it was confirmed that all linkages had been assigned a decision, and there were no more records to check.

Further checks were made on:

1. Private hospital births should not have a link to an HES delivery record unless the mother or baby was later transferred to an NHS hospital for any reason.
2. Each HES delivery record should link to the correct number of ONS birth records depending if it is for a singleton or multiple delivery.
3. The ONS birth record baby date of birth fell within the HES delivery admission period (no admission dates for home births). This was not always the case even when it was a confident link.
4. If there was more than one correct and relevant linked HES delivery record for an ONS birth record, any missing onset of delivery and method of delivery information was filled in from these.

## RESULTS

The quality assurance procedure took the equivalent of 55 working days for singleton births and 16 days for multiple births. The database software and processing limitations of the VML at that time slowed the procedure down. SQL Server has since been made available in VML and would speed this up considerably.

Seventeen per cent of singleton birth linked records were dealt with by the first two cleaning rules, and a further 80% were dealt with by rules 3 and 4. All the remaining rules were required to deal with the residual 3% of more complex cases.

Five per cent of multiple birth linked records were dealt with by the first two cleaning rules, and a further 60% were dealt with by rules 3 and 4. This left 35% as residual records, reflecting the greater complexity in multiple birth records.

For each year, the complex residual cases were clustered within particular trusts. This was probably due to information technology problems and changes to systems.

A summary of the results for singleton and multiple births is given in table 6A,B, broken down by calendar year.

Of the total 6 468 586 singleton ONS birth records, 97% had been linked to one or more HES delivery records and could be quality assured. Of these, 98% remained linked to one HES delivery record and 2% had all HES links discarded after quality assurance. Ninety-five per cent of all ONS singleton birth records overall were left with a link to an HES delivery record for analysis.

Of the 208 326 multiple births, 95% were linked to one or more HES delivery records and could be quality assured. Of these, 98% remained linked to one HES delivery record and 2% had all HES links discarded after quality assurance. Ninety-three per cent of all ONS multiple birth records overall were left with a link to an HES delivery record for analysis.

For singleton births, the percentage of all birth records left linked to one HES delivery record after quality assurance increased more or less year on year from 93% in 2005 to 95% in 2014. For multiple births, the rate increased from 94% in 2005 to 97% in 2010, but then started to decrease again to 87% in 2014. This reflects the same patterns as the percentage of ONS birth records that were originally linked to an HES delivery record and are biased by the number of records that linked to a HESID but no delivery records that were excluded from quality assurance.

### Reasons for discarding HES records linked to an ONS birth

After quality assurance, all or some of the available linked HES delivery records will have been discarded, leaving only a link to one correct HES record, or to none at all. Discarding all the linked HES records for a birth suggests incorrect linkage and is discussed in the next section. Discarding all but one is due to a variety of reasons why there may be more than one HES delivery record available for a birth.

Seventy-six per cent of all the linked HES delivery records that were discarded were due to duplicate copies of HES delivery episodes, and 0.5% were due to invalid delivery records, dealt with in the cleaning stages 1 and 2 of the methodology. A summary of the remaining reasons by year for singleton births is given in table 7.

'Same mother different baby' (column A) accounted for 60% of all discarded links. This is expected when the study period covers 10 years and each birth is linked by the mothers' identifiers to all other deliveries to that mother in that period. The second main reason which accounted for 21% of discarded links was when there were additional episodes relating to a delivery, usually a predelivery or postdelivery admission (column B). The numbers of these generally decreased over time, suggesting data quality is a factor. The third main reason, accounting for 16%, was when links to other episodes in a spell have been discarded, keeping only the link to the episode with the most amount of delivery information (column C).

The remaining three reasons each accounted for fewer than 2% of discarded links. Column D is when multiple delivery records contained the same information and links to superfluous records were discarded. This is also a data quality issue. There was a large increase in these in 2014. Column F relates to a small number of linked singleton birth records that were identified as 'potential multiple births', described later in this paper.

Column E is 'wrong links'; links discarded where there are no matching variables or insufficient numbers of matching variables according to the quality assurance methodology, and the baby's date of birth did not indicate a 'same mother different baby'. This group is of interest because it could be indicating erroneous false-positive links by the linkage algorithm, or it could indicate erroneous false negatives in the quality assurance methodology.

Fifty-seven per cent (2743) of this group was linked by match rank 1 of the linkage algorithm, the most reliable stage of the linkage algorithm that matches all available identifiers including the NHS number, suggesting they should be correct maternal linkages. These were checked in greater detail clerically for explanations.

Of these, 2323 (85%) were true wrong links where either none of the five baby data items matched at all, or only the place of birth matched, or only sex matched, or

**Table 6** Summary of quality assurance results

**(A) Singleton births by calendar year**

| Year | Total singleton ONS birth records | Total singleton ONS births linked to any HES delivery records by NHS Digital | % Total singleton ONS births linked to any HES delivery records by NHS Digital | Of singleton ONS births linked to any HES delivery records by NHS Digital, number left with link to one HES delivery record after QA | Of singleton ONS births linked to any HES delivery records by NHS Digital, % left with link to one HES delivery record after QA | Of singleton ONS births linked to any HES delivery records by NHS Digital, number left with no links to HES delivery records after QA | Of singleton ONS births linked to any HES delivery records by NHS Digital, % left with no links to HES delivery records after QA | % of *all* total singleton ONS births left with link to one HES delivery record after QA | % of *all* total singleton ONS births left with no links to HES delivery records after QA |
|---|---|---|---|---|---|---|---|---|---|
| 2005 | 599 237 | 565 559 | 94.4 | 554 566 | 98.1 | 10993 | 1.9 | 92.5 | 7.5 |
| 2006 | 620 730 | 589 127 | 94.9 | 573 770 | 97.4 | 15357 | 2.6 | 92.4 | 7.6 |
| 2007 | 638 995 | 612 782 | 95.9 | 595 585 | 97.2 | 17197 | 2.8 | 93.2 | 6.8 |
| 2008 | 656 196 | 635 411 | 96.8 | 621 006 | 97.7 | 14405 | 2.3 | 94.6 | 5.4 |
| 2009 | 653 322 | 636 284 | 97.4 | 621 423 | 97.7 | 14861 | 2.3 | 95.1 | 4.9 |
| 2010 | 665 599 | 652 533 | 98.0 | 641 167 | 98.3 | 11366 | 1.7 | 96.3 | 3.7 |
| 2011 | 666 582 | 653 552 | 98.0 | 642 263 | 98.3 | 11289 | 1.7 | 96.4 | 3.6 |
| 2012 | 676 399 | 661 511 | 97.8 | 648 501 | 98.0 | 13010 | 2.0 | 95.9 | 4.1 |
| 2013 | 647 666 | 633 222 | 97.8 | 622 943 | 98.4 | 10279 | 1.6 | 96.2 | 3.8 |
| 2014 | 643 860 | 628 032 | 97.5 | 617 263 | 98.3 | 10769 | 1.7 | 95.9 | 4.1 |
| Total | 6 468 586 | 6 268 013 | 96.9 | 6 138 487 | 97.9 | 129526 | 2.1 | 94.9 | 5.1 |

**(B) Multiple births by calendar year**

| Year | Total multiple ONS birth records | Total multiple ONS births linked to any HES delivery records by NHS Digital | % Total multiple ONS births linked to any HES delivery records by NHS Digital | Of multiple ONS births linked to any HES delivery records by NHS Digital, number left with link to one HES delivery record after QA | Of multiple ONS births linked to any HES delivery records by NHS Digital, % left with link to one HES delivery record after QA | Of multiple ONS births linked to any HES delivery records by NHS Digital, number left with no links to HES delivery records after QA | Of multiple ONS births linked to any HES delivery records by NHS Digital, % left with no links to HES delivery records after QA | % of *all* total multiple ONS births left with link to one HES delivery record after QA | % of *all* total multiple ONS births left with no links to HES delivery records after QA |
|---|---|---|---|---|---|---|---|---|---|
| 2005 | 18376 | 17404 | 94.7 | 17209 | 98.9 | 195 | 1.1 | 93.6 | 1.1 |
| 2006 | 19541 | 18522 | 94.8 | 18258 | 98.6 | 264 | 1.4 | 93.4 | 1.4 |
| 2007 | 20066 | 19257 | 96.0 | 18957 | 98.4 | 300 | 1.6 | 94.5 | 1.5 |
| 2008 | 20803 | 20100 | 96.6 | 19894 | 99.0 | 206 | 1.0 | 95.6 | 1.0 |
| 2009 | 22008 | 21338 | 97.0 | 21085 | 98.8 | 253 | 1.2 | 95.8 | 1.1 |
| 2010 | 21501 | 21033 | 97.8 | 20847 | 99.1 | 186 | 0.9 | 97.0 | 0.9 |
| 2011 | 22099 | 21199 | 95.9 | 20872 | 98.5 | 327 | 1.5 | 94.4 | 1.5 |
| 2012 | 22058 | 20166 | 91.4 | 19554 | 97.0 | 612 | 3.0 | 88.6 | 2.8 |
| 2013 | 20767 | 18735 | 90.2 | 18165 | 97.0 | 570 | 3.0 | 87.5 | 2.7 |
| 2014 | 21107 | 19015 | 90.1 | 18429 | 96.9 | 586 | 3.1 | 87.3 | 2.8 |
| All years | 208326 | 196769 | 94.5 | 193270 | 98.2 | 3499 | 1.8 | 92.8 | 1.7 |

HES, Hospital Episode Statistics; ONS, Office for National Statistics; QA, quality assurance.

**Table 7** Breakdown of reasons for discarding HES delivery records linked to an ONS birth excluding duplicate copies of HES delivery episodes and invalid deliveries for England singleton births by year

| Year | All<br>Total links broken (excluding exact duplicates and invalid records) | A<br>Same mother different baby | B<br>Multiple episodes—pre/postdelivery admission | C<br>Multiple episodes—part of spell | D<br>Multiple episodes—exact duplicates with different epikeys | E<br>Wrong link | F<br>Potential multiple birth |
|---|---|---|---|---|---|---|---|
| 2005 | 46164 | 24457 | 14941 | 5805 | 104 | 817 | 40 |
| 2006 | 49706 | 25082 | 16565 | 6984 | 204 | 782 | 89 |
| 2007 | 48999 | 26304 | 13187 | 8301 | 323 | 799 | 85 |
| 2008 | 47748 | 26032 | 9466 | 11473 | 89 | 614 | 74 |
| 2009 | 40519 | 25688 | 6137 | 7680 | 283 | 673 | 58 |
| 2010 | 37007 | 24993 | 5176 | 5819 | 366 | 599 | 54 |
| 2011 | 37013 | 24679 | 6033 | 5367 | 741 | 145 | 48 |
| 2012 | 36176 | 24966 | 5899 | 4298 | 838 | 146 | 29 |
| 2013 | 34285 | 23071 | 5709 | 5018 | 281 | 169 | 37 |
| 2014 | 31690 | 21980 | 3662 | 3957 | 1863 | 110 | 118 |
| All years | 409307 | 247252 | 86775 | 64702 | 5092 | 4854 | 632 |
| Percentage | 100 | 60.41 | 21.2 | 15.81 | 1.24 | 1.19 | 0.15 |

HES, Hospital Episode Statistics; ONS, Office for National Statistics.

only gestation matched. The baby's date of birth had not been transposed nor was 'same mother different baby', suggesting incorrect maternal linkage or some other data quality issue. This is 0.04% of all singleton birth records originally linked to an HES delivery record.

It appeared that 366 (13%) were potential false negatives made in error by the quality assurance methodology. The dates of birth in these do not match due to transposition in the date of birth elements. If this had been factored into the methodology, these links would have not been discarded.

The remaining 94 (3%) were undeterminable, with some discriminatory baby data items matching and some not. For example, the date of birth may match or have been transposed, but birth weight and/or place of birth may be very different.

Some of the categories of problem shown in table 7 may have been restricted to particular trusts but further investigation would be needed to check this.

### Differences between records with correct and incorrect linkage

Checks were carried out to ascertain if the ONS birth records that remained correctly linked to one HES delivery record after the quality assurance procedure, and the ONS birth records with all links to HES delivery records discarded after the quality assurance procedure differed significantly by key variables. Discarding all links could indicate linkage error, data quality issues or false negatives in the quality assurance methodology.

Online supplementary appendix B gives tables to compare the proportions of values for the key variables for both groups, and $X^2$ test results for testing the null hypothesis that the distribution of a variable is the same in the two groups. Variables from the ONS birth record include the match rank of the linkage algorithm, the region of birth, the year of birth, the month of birth, the day of birth (8) based on an 8-day categorisation including public holidays, the day of birth (11) based on an 11-day categorisation including specific public holidays and the days before and after them, the hour of birth, the age of the mother, the sex of the baby, the ethnicity of the baby, gestational age, if a stillbirth, and the location of birth. Missing gestational age and missing birth weight in HES were also included.

The null hypothesis is rejected for all variables except for sex, day (8) and day (11) in multiple births. Finding significant differences is to be expected because the data set is large and the $X^2$ test can pick up subtle differences, but this does not tell us very much. Comparing the difference in the percentage distribution of values for each variable for the two groups provides more information on how they differ. This is given in table 8 for singleton and multiple births where the difference is greater than 1%.

These differences give an indication of when incorrect linkage is more likely to occur, or indeed false negatives in the quality assurance. Differences between singleton

**Table 8** Comparison of differences in the percentage distributions of values for key variables for ONS birth records that remained correctly linked to one HES delivery record after the quality assurance procedure, and the ONS birth records with all links to HES discarded after the quality assurance procedure, for singleton and multiple births

| Variable | Singleton births | | Multiple births | |
| --- | --- | --- | --- | --- |
| | Value | Difference in percentages* | Value | Difference in percentages* |
| Match rank of linkage algorithm | 6 | 5.4 | | |
| Region of birth | Elsewhere | 2.3 | East of England | 1.9 |
| | Home | 38.3 | London | 5.8 |
| | | | Home | 2.8 |
| Year of birth | 2006 | 2.5 | 2012 | 7.4 |
| | 2007 | 3.6 | 2013 | 6.9 |
| | 2009 | 1.4 | 2014 | 7.2 |
| Month of birth | March | 2.0 | March | 1.7 |
| | | | April | 1.5 |
| Day of birth (8) | | | Saturday | 1.2 |
| Day of birth (11) | | | Saturday | 1.2 |
| Hour of birth | 3.00–5.59 | 1.7 | 6.00–8.59 | 1.4 |
| | 6.00–8.59 | 1.8 | 9.00–11.59 | 1.5 |
| Sex of baby | | | Male | 1.1 |
| Age of mother | 30–34 | 1.7 | 30–34 | 1.2 |
| | 35–39 | 1.3 | 35–39 | 2.4 |
| | | | 40–44 | 3.9 |
| Ethnicity of baby | Not known | 3.4 | Black African | 1.9 |
| | | | Not known | 1.4 |
| Gestational age | Preterm | 1.1 | Preterm | 1.1 |
| Stillbirth | Yes | 2.3 | Yes | 2.5 |
| Gestational age missing on HES | Yes | 20.1 | Yes | 10.9 |
| Birth weight missing on HES | Yes | 12.3 | Yes | 7.0 |
| Trust | PPL (Portland) | 1.7 | PPL (Portland) | 2.5 |
| | RGQ (Ipswich) | 2.0 | RGQ (Ipswich) | 1.5 |
| | RKE (Whittington) | 4.0 | RJ6 (Croydon) | 1.3 |
| | RVL (Barnet and Chase Farm) | 1.4 | RKE (Whittington) | 3.9 |
| | | | RQ8 (Mid Essex) | 1.6 |
| | | | RTH (Oxford University Hospitals) | 1.6 |
| | | | RVL (Barnet and Chase Farm) | 1.4 |
| | | | RVV (East Kent) | 2.0 |
| | | | RW3 (Central Manchester) | 1.1 |
| | | | RWH (East and North Hertfordshire) | 1.1 |

*Over 1%.
HES, Hospital Episode Statistics; ONS, Office for National Statistics.

and multiple births are seen. Some of these associations with incorrect linkage make sense. Home births are less likely to have an HES delivery record as many are not captured by hospital-based clinical systems from which HES delivery records are derived. Match rank 6 is a less reliable stage of the linkage algorithm because it does not match on the NHS number. HES data quality is previously known to be poorer in years 2006 and 2007, although for multiple births, years 2012–2014 are more likely to be incorrectly linked. Missing data for gestational age and birth weight in HES values means there is less information to make a correct link. The other factors require more investigation

or could be simply due to chance with the large numbers involved.

Online supplementary appendix table B.1 breaks down the proportions of each group by the match rank of the NHS Digital linkage algorithm, providing useful information about the quality of their linkage at each stage.

### Data and linkage quality issues

Quality assurance of the linked data set for such a large population revealed issues with the quality of the data within each of the data sets and also the quality of linkage that otherwise might not be detectable. While it is already known that HES has data quality issues, issues were found in the ONS birth records that warranted further investigation.

First, the consistent common location code that was assigned to the place of birth codes was found to not match between birth registration and notification for 22% of the singleton births with a correct HES delivery link after quality assurance. This suggested the linkage between birth registration and notification could be incorrect. A sample of these records that also had very discordant mother and baby identifiers and baby tail information was given to ONS, which carried out this linkage, for checking.

ONS confirmed that the baby NHS number or the sequence number[18] matched for these cases and therefore the linkage was correct between ONS birth registration and birth notification, regardless of the discordant variables. However, no quality assessment of the sequence number assignment process was available, so 542 of the most suspect cases were flagged. It was decided to exclude these from analyses.

Second, a small number of singleton birth records appeared to be multiple birth records (see table 4 column F) based on a combination of HES variables suggesting a multiple birth delivery, and other birth registrations existing to the same mother on the same day, yet these were not labelled as a multiple birth by ONS. Again, these cases were checked by ONS which confirmed that they should be multiple births. The error was explained by a delay in some records being registered and received, possibly causing issues with finding the corresponding record in its system.

Lastly, after quality assurance, a small number of multiple births did not have all the expected number of ONS birth records, for example, a twin birth only had one ONS birth record associated with it rather than two. These were not always explained by stillbirths prior to 24 weeks' gestation which should not be registered as a birth. Investigation revealed that the missing ONS birth records could often be found in the 'unlinked' ONS birth records file, to which no HES records were linked. This could be the case even if the mothers' identifiers, including NHS number, clearly matched.

This was discussed with our contact at NHS Digital who investigated this. It was explained that these missed links were due either to the HES Patient Index having been updated since the linkage was done originally, or linkage error. The project did not attempt to find any further missing links.

The advantage of having the identifiers available to the project team for both ONS birth and HES delivery records made it possible to investigate and explain the above issues, but other research projects may not. Overall, this highlights how researchers and analysts should not assume that any routinely collected administrative data sets and linkages carried out between them are error free.

## DISCUSSION
### Main findings

This paper describes the quality assurance process that was carried out on 10 years of linked ONS birth records and Maternity HES delivery records at a national level using patient identifiers. The aim was to prepare the linked data set for analysis but the results also shed light on the quality of the data and the linkage itself. This makes it informative for other users of similarly linked data and for NHS Digital staff doing the linkage.

The method adopted compares common baby data items, giving most importance to matching babies' date of birth and location of birth. Altogether, 95% of all singleton births and 93% of all multiple births in England in the period 2005–2014 were linked to one correct HES delivery record after quality assurance.

Administrative duplicate HES records were found to be the main reason for discarding links. Other reasons were due to other births to the same mother within the study period, the presence of multiple HES records per delivery for various reasons, and error in the linkage and in the quality assurance method itself.

Certain features were more common in incorrectly linked records than correctly linked records. These included births linked using match rank 6 of the linkage algorithm, births at home and in the London region, and births where the birth weight or gestational age was missing in HES. This suggests some bias in the linkage and often relates to data quality issues. These patterns differed between singleton and multiple births.

Missed links were also discovered, which together with the previous findings provide further evidence on the quality of the linkage.

It was possible to identify aspects of the quality of the ONS birth records data and have ONS investigate them.

### Strengths and limitations, and future work

The quality assurance procedure was able to draw upon the availability of identifiers in both the ONS and HES data to inform the choice of rules at each stage in the methodology, to carry out clerical checks to ensure the accuracy of the results, and to explore data and linkage quality issues. The procedure successfully identified correct and incorrect linkages to prepare the large linked data set for analysis within the resources available.

Because personally identifiable data were used, the work was carried out in the secure environment of ONS' VML. While security was safeguarded, this led to many delays and disruption due to software and processing

limitations of systems and servers. This was costly in terms of researcher time and has limited the extent of analytical outputs from the study.

Quality assurance could only be carried out on ONS birth records that had linked to an HES delivery record. Links where a HESID was found but no delivery record was available were excluded. If the main reason for this is because some trusts did not submit the HES delivery record, then bias will be inherently present in which linked records could be quality assured and which could not.

The methodology to decide if the linked records related to the same baby was purely deterministic and did not allow for transposition of date of birth elements, or for rounding and truncating in the birth weight and gestational age values. It did not strongly discriminate between when values were missing rather than different.

A disproportionate amount of effort was spent on the approximately 4% of complex linked records that remained after stages 1–4 of the methodology.

Future work could address these issues by incorporating more flexible matching criteria and more thoroughly preparing the birth weight and gestational age variables for data entry errors beforehand. Ways to simplify the methodology to fewer variable combinations or rules to identify correct and incorrect links could be explored. Future work could also assess how variation in data quality at trust level affects linkage quality and the results.

## Implications

The linkage error identified by quality assuring the linked data set suggests that linkage carried out using similar algorithms and by 'Trusted Third Parties' generally, cannot be assumed to be error free and may affect any analyses carried out on them. This has implications for other research projects that are not informed of the quality of the linkage, or have the means to quality assure it themselves using identifiers. In particular, no assessment of the quality of the linkage algorithm is routinely published by NHS Digital.

A very similar linkage algorithm is used to routinely link HES data to ONS death registration data, differing only at match rank 8,[19] although different approaches were used earlier by the National Centre for Health Outcomes Development[20] and the Department of Health. These linked data are used to analyse outcomes of hospital care such as immediate and long-term survival rates and hospital-based mortality indices including the Standard Hospital Mortality Index (SHMI) and indices of 30-day mortality. Concerns over how quality of this linkage can affect analyses using the 30-day mortality indicator have been reported.[21] On the other hand, an evaluation of the mortality linkage for SHMI purposes found that 'the proportion of erroneous linkages is sufficiently small to not impact on a SHMI result,' but its report did not explicitly describe the linkage algorithm used.[22] The relevant member of its study team has confirmed that the team did not know what linkage algorithm was used by the then Information Centre for Health and Social Care

to link the mortality data for SHMI purposes, and they were not provided with NHS numbers or other identifiable variables to enable performing the linkage themselves or indeed to investigate the quality of the linkage (Fotheringham J, personal communication).

Linkage of identifiable data for research purposes is mostly carried out by 'Trusted Third Parties' such as NHS Digital. The Trusted Third Party model replaces identifiers with an encrypted code, and only the de-identified data are made available to researchers in a secure environment. The Digital Economy Bill[23] announced in May 2016 sought increased access to de-identified data for research, meaning that reliance on Trusted Third Party linkage would increase. The Bill implies that data accessed under the new legislation must only be linked using the Trusted Third Party model. However, responses to this and to The Cabinet Office consultation document 'Better Use of Data in Government'[24] from bodies such as the Royal Statistical Society[25] and the Health Statistics User Group[25] have expressed concern over this and also that the proposed legislation does not include health and social care data. Elements of the Digital Economy Bill could be interpreted as allowing flexibility in who can access and link identifiable data so long as disclosure is avoided and other conditions and codes of practice are in place. Clarification on this is required so that it can be established if researchers will be restricted only to pre-linked data without any means to assess their quality.

In the context of the use of linked data for statistics, the ONS has expressed a preference to be able to link data with identifiers rather than encryption to achieve better quality official population statistics in the future.[26]

## CONCLUSION

Carrying out quality assurance of linked records has informed knowledge of both the quality of the data and of the linkage, providing benefits over cleaning each data set in isolation. Other research projects using large linked routinely collected data sets should not assume the linked data sets are error free or optimised for their analysis. This has implications especially for users of anonymised linked data. The resources needed to do quality assurance and prepare effectively prior to generating descriptive or analytic reports are considerable and need to be addressed when planning and costing large data linkage projects.

**Acknowledgements** The author thanks the members of the project team and its Study Advisory Group for their help and advice, in particular Nirupa Dattani who managed the linkage, Rod Gibson who set up the database and Alison Macfarlane as principal investigator. The author also thanks all the relevant colleagues in the Office for National Statistics and NHS Digital, formerly the Health and Social Care Information Centre for their help, in particular Emma Gordon, Joanne Evans, Claudia Wells, Alex Lloyd, Justine Pooley, Elizabeth Mclaren and members of the VML Team at the Office for National Statistics, and Ariane Alamdari and Garry Coleman at NHS Digital.

**Contributors** GH was responsible for the design of the quality assurance methodology, writing all parts of the paper, and giving final approval of the version to be published. GH agreed to be accountable for all aspects of the work in ensuring that questions related to the accuracy or integrity of any part of the work are appropriately investigated and resolved.

**Funding** This work was part of a project funded by the HS&DR Programme (project number 12/136/93) and will be published in full in the Health Services & Delivery Research Journal. Further information available at: https://www.journalslibrary.nihr.ac.uk/programmes/hsdr/1213693/#/

**Disclaimer** This report presents independent research commissioned by the National Institute for Health Research (NIHR). The views and opinions expressed by authors in this publication are those of the authors and do not necessarily reflect those of the NHS, the NIHR, MRC, CCF, NETSCC, the Health Services & Delivery Research programme or the Department of Health. The data were processed in the secure environment of the Office for National Statistics' Virtual Microdata Laboratory and the following disclaimer applies: This work contains statistical data from ONS which is Crown Copyright. The use of the ONS statistical data in this work does not imply the endorsement of the ONS in relation to the interpretation or analysis of the statistical data. This work uses research data sets which may not exactly reproduce National Statistics aggregates.

**Competing interests** None declared.

**Patient consent** Not required.

**Ethics approval** Ethics approval 05/Q0603/108 and subsequent substantial amendments were granted by East London and City Local Research Ethics Committee 1 and its successors. Permission to use patient identifiable data without consent under Section 60 of the Health and Social Care Act 2001 was initially granted by the Patient Information Advisory Group PIAG 2-10(g)/2005. Renewals and amendments and a second permission CAG 9-08(b)2014 under Section 251 of the Health Service (Control of Patient Information) Regulations 2002 were granted by its successor bodies, the Ethics and Confidentiality Committee of the National Information Governance Board and the Confidentiality Advisory Group of the Health Research Authority. A second permission CAG 9-08(b)2014 to use patient identifiable data without consent under Section 251 of the Health Service (Control of Patient Information) Regulations 2002 and create a research database held at the Office for National Statistics for analyses relating to inequalities in the outcome of pregnancy and to inform maternity service users about the outcome of midwifery, obstetric and neonatal care was granted by the Confidentiality Advisory Group of the Health Research Authority. Permission to access data from the Office for National Statistics in the VML was granted by ONS' Microdata Release Panel. All members of the research team successfully applied for ONS Approved Researcher Status. Permission to link and analyse data held by the Health and Social Care Information Centre, now NHS Digital, was granted under Data Sharing Agreement NIC-273840-N0N0N.

**Provenance and peer review** Not commissioned; externally peer reviewed.

**Data sharing statement** The author does not have permission to supply data or identifiable information to third parties, including other researchers, but the research team of which she is a member has permission under Section 251 of the Health Service (Control of Patient Information) Regulations 2002 to analyse patient identifiable data for England and Wales without consent and create a research database which could be accessed by other researchers using the VML at the Office for National Statistics subject to permission from the Confidentiality Advisory Group of the Health Research Authority to access patient identifiable data without consent.

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
