## [Reviewer comments · BMJ Open]

ARTICLE DETAILS

TITLE (PROVISIONAL)	Linkage of Maternity Hospital Episode Statistics Data to Birth Registration and Notification Records for Births in England 2005-2014: 2 - Quality Assurance of Linkage of Routine Data for Singleton and Multiple Births
AUTHORS	Harper, Gillian

VERSION 1 – REVIEW

REVIEWER	Rachael Wood NHS National Services Scotland Information Services Division Scotland UK I am responsible for maintaining linkage of maternal and baby NHS records in Scotland but have had no involvement in this English linkage.
REVIEW RETURNED	10-Jun-2017

GENERAL COMMENTS	This paper reports on the quality assurance of the linkage of birth statutory registration and NHS notification records to HES delivery records. All births in England 2005-2014 are included. This is painstaking work that has resulted in a cleaned linked dataset being available to the research community through ONS – and as such it should be commended. The very considerable amount of work involved is clear. Such detailed scrutiny of linkage quality, and associated transparent reporting, is relatively uncommon and again the author should be commended for putting this information into the public domain. I concur with the conclusion that accuracy and representativeness of large linked datasets presented to researchers by ‘trusted third party’ linkers should not be taken for granted – the potential for differential linkage completeness in certain groups introducing bias should always be considered. In practice it is not – either through naive acceptance, or because relevant information is not included in the anonymised linked dataset provided to researchers. Finding ways for researchers and TTP linkers to work together to enable transparent scrutiny of linkage quality whilst maintaining individuals’ privacy is an urgent priority. The paper highlights known data quality issues within HES maternity data. Perhaps more worryingly, it also suggests data quality issues within the ‘gold standard’ statutory birth registration data. This warrants further investigation. Unusually I have no substantive points of clarification or corrections relating to the paper. In my view it can be published ‘as is’.
---

REVIEWER	Toan C Ong University of Colorado Anschutz Medical Campus, Aurora, CO United States of America
REVIEW RETURNED	08-Jul-2017

GENERAL COMMENTS	Thank you for the opportunity to review this paper which reports the results of the quality assurance process of linkage of birth data between the Office for National Statistics (ONS) and the Maternity Episode Statistics (HES) in England. This paper attempted to verify both linkages of singletons and multiples. It is important to note that linking multiples is one of the most difficult challenges in records due to the similarity of linkage value between these records. The efforts by the author to record and report the results are remarkable. Having extensive data included in the paper made it easy to find an explanation for a question. Singleton linkage and multiple births linkage are often done together. So the separation and comparison of these two different cohorts in this paper is very much appreciated by this reviewer. Several comments:  • Overall, the paper was not well organized. Discussions in each section are fragmented and hard to follow. The author uses multiple single-sentence paragraphs which are rarely appropriate to articulate an idea in scientific papers. Each paragraph should start with a main point, followed by the evidences to support that point. • Is the NHS number a reliable linkage variable? • On page #8, line 17-24: the relationship between the two paragraphs in this section is unclear. My interpretation is that paragraph #1 raised the issue about the challenge to verify multiple births linkage because each ONS birth record got linked to one's siblings of the same delivery. Paragraph #2 provides a solution in which date of birth, sex, gestational age, birthweight and location birth are used to make sure that the same records are linked correctly. Among these variables, only birthweights can be used to distinguish babies in multiple birth (e.g., twins, triplets). If birthweight is missing or inaccurate, what is the alternative solution? From the linkage method, it is not clear how twins and triplets were linked. • It is clear that there are a lot of errors produced by the linkage method due to multiple different reasons. Carrying out clerical review process for every linkage result is possible not scalable. Problems with data quality can improved but cannot be completely avoided. How will the findings of this paper inform the improvement of the linkage method? • The hypotheses in the Results section should be stated stated formally. • The author should consider moving the data quality discussion to the "data sources" section. • Page #11, line 12-18, that paragraph is redundant as it was discussed in previous sections. • Page 8, line 26: There is an error message: "Error! Bookmark not defined" • This paper will benefit from extensive copyediting.
---

VERSION 1 – AUTHOR RESPONSE

Reviewer 1:

Thank you for your comments.

Reviewer 2

Thank you for your comments. I have added my responses to each of your points below.

Comment: Overall, the paper was not well organized. Discussions in each section are fragmented and hard to follow. The author uses multiple single-sentence paragraphs which are rarely appropriate to articulate an idea in scientific papers. Each paragraph should start with a main point, followed by the evidences to support that point.

Response: I have reviewed this and have maintained the single sentence paragraphs. This is because they relate to a single stand-alone point each time, and are used mainly to describe part of the methodology etc. and not to articulate an idea with evidence.

Comment: Is the NHS number a reliable linkage variable?

Response: It is a unique person identifier. Linkage variables are discussed in N Dattani's preceding paper (paper 1). Paper 1 points out that errors can be made in recording it in some administrative systems. This is why linkage did not rely on the NHS number alone.

Comment: On page #8, line 17-24: the relationship between the two paragraphs in this section is unclear. My interpretation is that paragraph #1 raised the issue about the challenge to verify multiple births linkage because each ONS birth record got linked to one's siblings of the same delivery. Paragraph #2 provides a solution in which date of birth, sex, gestational age, birthweight and location birth are used to make sure that the same records are linked correctly. Among these variables, only birthweights can be used to distinguish babies in multiple birth (e.g., twins, triplets). If birthweight is missing or inaccurate, what is the alternative solution? From the linkage method, it is not clear how twins and triplets were linked.

Response: I have moved the second paragraph you refer to, to before Box 4 on page 9, and re-worded it to give more information on how multiple births were dealt with i.e. by using the birth order field to help distinguish between multiple babies. Detail is given in Appendix A.

Comment: It is clear that there are a lot of errors produced by the linkage method due to multiple different reasons. Carrying out clerical review process for every linkage result is possible not scalable. Problems with data quality can improved but cannot be completely avoided. How will the findings of this paper inform the improvement of the linkage method?

Response: There weren't 'a lot' of errors per se. It was more that because the linkage had to be for the mother and due to the nature of the HES data, lots of surplus links were also made that had to be identified and removed where necessary to leave only one link for each birth. I state in the discussion that the results have been fed back to NHS Digital, and that future work should look at trust level issues which are responsible for most of the data quality issues. Being aware of these beforehand would minimise the clerical checking.

Comment: The hypotheses in the Results section should be stated formally.

Response: There is no hypothesis to state

Comment: The author should consider moving the data quality discussion to the "data sources" section.

Response: Thanks for the suggestion. I have considered and decided to leave it where it is.

Comment: Page #11, line 12-18, that paragraph is redundant as it was discussed in previous sections.

Response: I agree, but that lines 12 to 15 only can be discarded. Lines 15 to 18 are needed

Comment: Page 8, line 26: There is an error message: "Error! Bookmark not defined"

Response: Fixed.

Comment: This paper will benefit from extensive copyediting.

Response: Extensive copyediting already been done to find balance between the technical content and plain English. No more will be done, especially in light of the first reviewer not finding it necessary

VERSION 2 – REVIEW

REVIEWER	Toan C Ong University of Colorado Denver
REVIEW RETURNED	05-Sep-2017
GENERAL COMMENTS	Based upon reviewing the document titled "GHarper_QA_Procedure_BMJOOpen_RevisedJuly2017_Revised_manuscript_marked_copy", the changes made to the initial submission were very limited. Most of this reviewer's questions were not answered. The author should provide a "Response to reviewers" which includes point-to-point response to reviewers' comments/questions.